# MUA-RL: Multi-turn User-interacting Agent Reinforcement Learning for agentic tool use

## Abstract

Recent advances in Agentic Intelligence have highlighted the importance of agentic tool use in Large Language Models (LLMs), especially when interacting with users. During multi-turn interactions, the dynamic, uncertain, and stochastic nature of user demands challenges agents to iteratively refine their understanding of user needs through communication while invoking tools to resolve queries, rather than simply calling tools for results. Existing reinforcement learning (RL) approaches for tool use lack the integration of genuinely dynamic users during the RL training process. To bridge this gap, we introduce MUA-RL (Multi-turn User-interacting Agent Reinforcement Learning for agentic tool use), a novel reinforcement learning framework that, for the first time in the field of agentic tool use, integrates LLM-simulated users into the reinforcement learning loop. MUA-RL aims to enable autonomous learning of models to communicate with users efficiently and use various tools to solve practical problems in dynamic multi-turn interactions. Evaluations on several benchmarks demonstrate that MUA-RL-32B outperforms or matches much larger open-source models such as DeepSeek-V3-0324 and Qwen3-235B-A22B in non-thinking setting (see Figure 1).

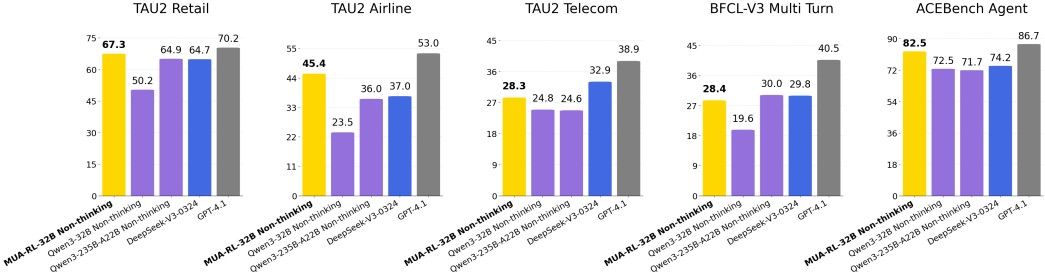

Figure 1: MUA-RL main results.

## 1 Introduction

The paradigm of LLMs is shifting towards Agentic Intelligence (Anthropic, 2025; Team et al., 2025a;b) – models are now equipped with extensive tools to interact with users and the world. This shift signifies a move away from static imitation learning, paving the way for models that engage in active learning through interactions, going beyond their pre-training and post-training data. Consequently, this new paradigm demands more on the model's ability to use tools effectively in multi-turn interactions and communicate clearly with users. While current LLMs mainly undergo supervised fine-tuning (SFT) on synthesized tool-using data to acquire capabilities to interact with both users and the world, RL is believed to have better generalization than SFT (Team et al., 2025a).

Combined with RL, LLMs have shown remarkable progress, particularly in static, well-defined domains. For example, DeepSeek-Math (Shao et al., 2024) achieves strong performance in mathematical problem solving. Recent works have begun to incorporate interactions with external environments into the RL training process. Retool (Feng et al., 2025) enhances RL by integrating code interpreter to enable LLMs to interact with a real code sandbox, SkyRL (Cao et al., 2025) facilitates

interactions between LLMs and actual Docker environments during RL training, and RAGEN (Wang et al., 2025) depends on symbolic or pre-specified environments. However, these systems generally operate within predetermined environments and rely on pre-scripted queries. As a result, current RL approaches may struggle on the dynamics and unpredictability of real-world interactions with users.

In real-world scenarios, users' needs are highly dynamic and unpredictable. Users often adjust their questions and expectations based on the model's responses, creating a feedback loop that requires continuous bidirectional adaptation. This interactive and co-evolutionary dynamic is currently overlooked in existing RL training frameworks and remains largely unexplored in practice.

To investigate the impact of dynamic user roles during the reinforcement learning rollouts, we propose MUA-RL, a novel reinforcement learning framework that, for the first time in the field of agentic tool use, integrates LLM-simulated users into the reinforcement learning loop. MUA-RL is designed to employ end-to-end reinforcement learning to enhance the agent's ability to iteratively refine its understanding of user intent through ongoing dialogue, while actively invoking tools to fulfill user requests. Unlike approaches that incentivize specific tool-calling formats or success rates at intermediate steps, our framework provides reward solely based on ultimate task completion. This encourages more bold and diverse exploration during reinforcement learning processes, fostering the emergence of more robust and general behavioral patterns. Ultimately, the agent autonomously evolves its capability to handle multi-turn tool-using tasks.

Our main contributions are as follows:

- We proposed MUA-RL, a novel multi-turn user-interacting reinforcement learning framework that incorporates LLM-simulated users into the reinforcement learning rollouts.

- We constructed two agentic data synthesis pipelines for high-quality cold-start: one with LLM-simulated tool responses, and another with real MCP server tool responses.

- We conducted detailed analysis of the model's training dynamics throughout the multi-turn user-interacting reinforcement learning process.

## 2 RELATED WORK

**Reinforcement learning.** Recent progress in RL for LLMs has moved beyond classical algorithmic frameworks toward more scalable and efficient optimization paradigms. Early work primarily leveraged PPO (Schulman et al., 2017) and actor–critic approaches (Haarnoja et al., 2018) to align models with human-preferred behaviors. Subsequent developments have introduced policy variants such as RLOO (Kool et al., 2019), GRPO (Shao et al., 2024), and DAPO (Yu et al., 2025), which aim to improve stability and sample efficiency under large-scale training. In parallel, STaR (Zelikman et al., 2022), through its iterative self-derivation process, and MCTS (Hao et al., 2023), via its lookahead-based exploration, have both demonstrated the ability to significantly reduce reliance on external supervision while substantially improving sample efficiency. More recently, RL has been integrated into broader LLM fine-tuning frameworks, enabling models to adapt flexibly to diverse downstream tasks such as search (Jin et al., 2025), coding (Liu & Zhang, 2025), and multimodal tasks (Shen et al., 2025). These advances underscore the versatility of RL as a general optimization framework for enhancing LLM capabilities, providing a foundation for extending RL techniques into new domains of model alignment and task specialization.

**Agentic tool use.** The development of tool use in LLMs has progressed through a series of methodological stages. Initially, prompting-based approaches relied on carefully designed instructions to elicit tool invocation without additional training (Chen et al., 2022; Lei et al., 2023). While lightweight and flexible, these methods often exhibited unstable tool behavior and struggled with complex tool interactions. Subsequently, SFT methods emerged, where models were trained on curated or distilled trajectories to improve their accuracy and consistency in tool use (Lin et al., 2024; Zhang et al., 2025a; Acikgoz et al., 2025). These approaches achieved more reliable performance but were constrained by the coverage and quality of training data. More recently, RL techniques have been employed to optimize tool-using policies through outcome-driven feedback (Song et al., 2025; Sun et al., 2025; Li et al., 2025b; Singh et al., 2025), allowing models to explore and refine tool invocation strategies in interactive environments. Despite these advances, most existing works

center on text-based or multi-step tasks (Section 3.3.1), and challenges remain in dynamic multi-turn user-interacting tool use.

# 3 METHODOLOGY

## 3.1 TASK FORMULATION

Multi-turn tool-using tasks involve dynamic interactions between the user and the agent and extensive exchanges between the database and the agent. At each interaction turn, the agent may: (a) invoke one or more tools sequentially to interact with the database for information retrieval or operations, *or* (b) communicate textually with the user to acquire information and discern their intent.

Formally, define a tuple $(\mathcal{T}, \mathcal{M}, \mathcal{O})$ where: $\mathcal{T}$ is the tool set space, $\mathcal{M}$ is the message space, $\mathcal{O} = \mathcal{O}_{db} \cup \mathcal{O}_{user}$ is the observation space, with $\mathcal{O}_{db}$ and $\mathcal{O}_{user}$ denoting the database and user observation subspaces respectively. Given a user query $o_{1,user} \in \mathcal{O}_{user}$, a typical multi-turn trajectory is expressed as:

$$\underbrace{(o_{1,\text{user}} \to t_1 \to o_{1,\text{db}} \to ... \to m_1)}_{\text{turn } 1}, ..., \underbrace{(o_{k,\text{user}} \to t_j \to o_{j,\text{db}} \to ... \to t_{j+j_k} \to o_{j+j_k,\text{db}} \to m_k)}_{\text{turn } k}, ... \quad (1)$$

where $t_i \in \mathcal{T}$ denotes invoking tool, $o_{i,db}$ denotes observation from database after invoking $t_i$, $o_{i,user}$ is the observation from the user, and $m_i \in \mathcal{M}$ represents the agent's message to the user. At each turn, the agent autonomously decides *whether* to invoke tools and *how many* tools to invoke, in order to make progress towards solving the user query.

## 3.2 AGENTIC DATA SYNTHESIS PIPELINE FOR HIGH-QUALITY COLD-START

The multi-turn tool-using tasks introduced in Section 3.1 present significant challenges for language models. The agent must autonomously invoke unfamiliar tools, and iteratively act through text-based communication, tool invocation, and error correction. To address these challenges, we employ a lightweight SFT phase (cold-start) to establish the basic capabilities of models in handling multi-turn tool-using tasks prior to deploying them within a RL framework for self-iteration.

Although the real world provides rich and authentic interaction scenarios, conducting data collection in such settings is difficult due to cost limitations, system complexity, privacy concerns, and accessibility barriers. Recent research efforts (Mitra et al., 2024; Guo et al., 2024; Team et al., 2025a; Sun et al., 2025) have focused on synthetic approaches for generating tool-using data, where large language models (LLMs) are typically employed to simulate tool execution results. Notably, the emergence of Model Context Protocol (MCP) (Hou et al., 2025) now offers an alternative approach, enabling direct utilization of accessible MCP servers for real tool execution.

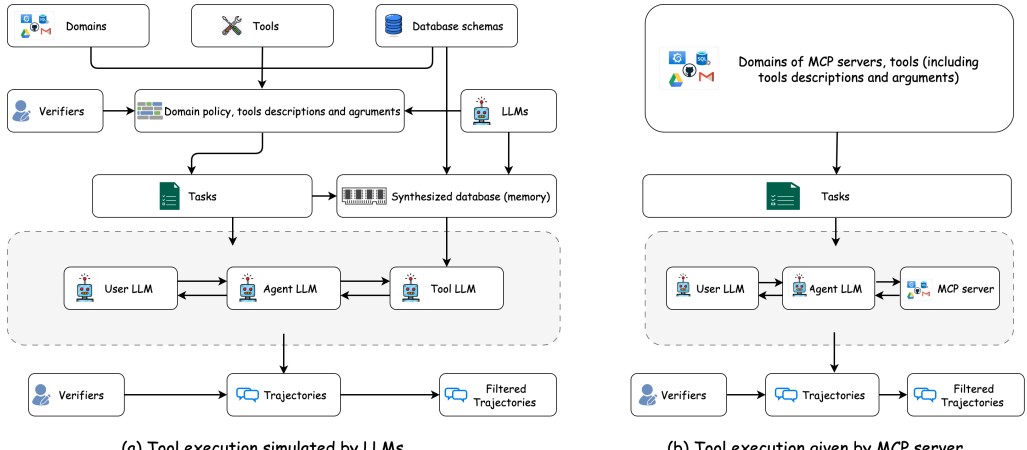

(a) Tool execution simulated by LLMs     (b) Tool execution given by MCP server

Figure 2: Agentic data synthesis pipeline. (a) Tool execution results simulated by LLMs. (b) Tool execution results given by real MCP server.

**Tool execution simulated by Large Language Models (LLMs).** In this scenario, we first come out a domain and its simplest possible database schemas, along with corresponding tools, inspired by (and simplified from) its real-world counterpart. Next, descriptions and arguments of the tools, and domain policy are generated by LLMs, followed by human curation and iterative refinement. Then, we generate trajectories through the collaboration of three LLMs: one serving as the *agent*, another as the *user*, and–most critically–one as the *tool*. While the roles of the agent and user are relatively straightforward, the tool LLM requires careful design to ensure the reliability of the whole generation process. Specifically, when constructing a domain-specific query, we employ LLMs to generate a small synthetic database (serving as memory) that conforms to the predefined database schemas. This memory is then provided to the tool LLM. During trajectory generation, when the agent LLM invokes a certain tool, it passes the tool name and corresponding arguments to the tool LLM. The tool LLM subsequently processes this input and generates an appropriate tool response based on the provided memory, as shown in Figure 2(a).

**Tool execution given by real Model Context Protocol (MCP) server.** In the MCP scenario, the process is significantly simplified. All tools are presented upfront, eliminating the need for manual design, and the MCP server automatically handles all tool execution. What remains is generating domain-specific user queries corresponding to the MCP server and producing trajectories through the interactions between the agent LLM, user LLM, and the MCP server, as illustrated in Figure 2(b).

It is noteworthy that all cold-start datasets undergo dual-verification, which combines human expert annotation with DeepSeek-R1 (Guo et al., 2025) evaluation, to filter out invalid trajectories and ensure data quality and validity. Trajectory examples are provided in Appendix C.

### 3.3 MUA-RL: Multi-turn User-interacting Agent Reinforcement Learning for Agentic Tool Use

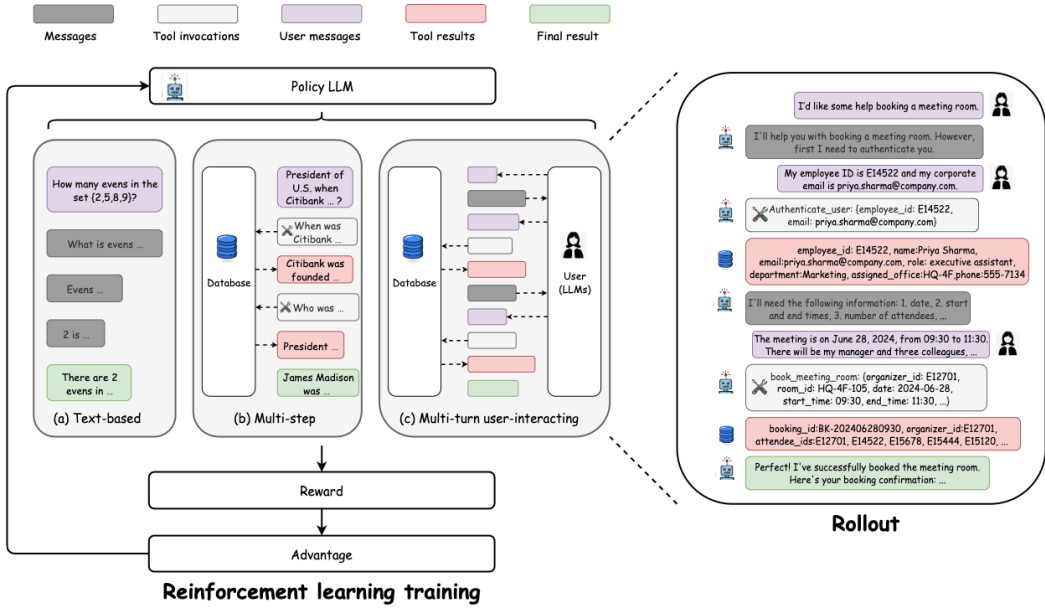

Figure 3: Three different kinds of the rollout processes. (a) The text-based rollout. (b) The multi-step rollout with tool execution. (c) The multi-turn user-interacting rollout with tool execution.

#### 3.3.1 Multi-turn User-interacting Rollout with Real-time Tool Execution

In conventional reinforcement learning rollouts, the policy LLMs typically perform only text-based generation, and then output a final answer to obtain a reward (e.g., in most mathematical reasoning tasks (Shao et al., 2024; Seed et al., 2025)). During inference, they do not use tools to interact with external systems–even commonly used ones like code interpreter (CI)–as illustrated in Figure 3(a).

Recent works explore integrating text generation with executable tools in rollouts. As demonstrated in Figure 3(b), these approaches dynamically interleave natural language generation by the policy LLM with real-time tool-execution results on the fly. Here, the term "database" can carry different meanings depending on the scenarios. With a CI, the "database" may denote a real code-execution sandbox environment (Li et al., 2025b; Feng et al., 2025), whereas in deep research settings it can refer to the internet as an information-retrieval source (Jin et al., 2025; Sun et al., 2025; Song et al., 2025; Chen et al., 2025b; Li et al., 2025a). This interactive rollout paradigm (also called multi-step rollout), which engages with real-world databases, enables the model to develop practical tool invocation capabilities for solving domain-specific problems, moving beyond pure textual generation.

Advancing further, recent works have explored using LLMs as simulators of human characters (Kim et al., 2022; Park et al., 2023), which opens new possibilities for integrating user roles in large-scale reinforcement learning rollout processes. Building upon this, in our reinforcement learning framework, we introduce automated users simulated by LLMs to interact with the agent during rollouts. Compared to the text-based rollout and multi-step rollout, the agent must not only communicate with the user via text to gather user information and discern user's intentions, but also utilize provided tools to interact with the database – retrieving information or performing operations to fulfill the user's requests. The complete rollout integrates multiple components: text-based generation, tool invocation, user messages, and tool-execution results, which significantly enhances the dynamics, stochasticity, and uncertainty of the rollout processes. Through this highly dynamic framework, we aim to train agents to autonomously develop more sophisticated behavioral patterns, requiring both appropriate tool usage and effective communication with users, as depicted in Figure 3(c).

### 3.3.2 REWARD

In existing reinforcement learning works on agentic tool use, the reward design is often quite complex. Since tool invocation is a highly structured behavior for language models, reward design typically incorporates format rewards (Singh et al., 2025; Qian et al., 2025; Feng et al., 2025), tool-name or parameter-name matching rewards (Qian et al., 2025; Zhang et al., 2025b), and tool execution rewards that measure fraction of successful tool calls (Singh et al., 2025; Zeng et al., 2025b). However, complex reward design may not enable models to learn effective behavioral patterns in dynamic multi-turn interactions. Instead, the model may becomes discouraged from trial and error. In our work, we simplify the reward design, the reward $r = 1$ only when the agent successfully fulfills the task in accordance with the system prompt, and $r = 0$ otherwise. This reward design offers two advantages. (1) **Robust to dialogue variation**: the evaluation is invariant to the specific conversational trajectory or tool invocation sequence, allowing for diverse behaviors as long as the correct outcome is achieved. (2) **Mitigation of reward hacking**: agents cannot directly exploit the output format or tool invocation syntax, but can only be rewarded by complete task resolution.

## 4 EXPERIMENTS

In this section, we demonstrate the superiority of MUA-RL in terms of performance, robustness, and generalization across multiple multi-turn tool-using benchmarks, and conduct in-depth analyses to verify the effectiveness of our multi-turn user-interacting reinforcement learning framework.

**Training.**  We select Qwen3 Non-thinking series (Yang et al., 2025) as the primary backbone models in our experiments. The training process consists of two stages. First, for cold-start phase, we fine-tune the models on trajectories synthesized by pipelines described in Section 3.2. Second, for RL phase, We implemented MUA-RL framework based on VolcEngine Reinforcement Learning (VeRL) (Sheng et al., 2024) and integrated a real, operational database environment for validating the results generated by tool invocation. 115 retail and 50 airline datasets from **TAU1-Bench** (Yao et al., 2024) are used as training data, with GPT-4o-2024-11-20 (Hurst et al., 2024) serving as the user simulator. GRPO (Shao et al., 2024) is adopted as RL training alghorithm. Further details are provided in Appendix B.2.

**Evaluation.**  We evaluate our approach on four representative multi-turn tool-using benchmarks. In addition to the previously introduced TAU1-Bench, three other multi-turn tool-using benchmarks – **TAU2-Bench** (Barres et al., 2025), **Berkeley Function-Calling Leaderboard (BFCL)-V3 Multi**

**Turn** (Patil et al., 2025), and **ACEBench Agent** (Chen et al., 2025a), are evaluated to demonstrate the generalization of MUA-RL. For TAU1-Bench, TAU2-Bench and ACEBench Agent, GPT-4.1 is used as the user simulator. For each test set, we conduct four repeated tests and take the average to improve confidence. All evaluations are conducted under a deterministic inference setting with temperature fixed to 0.0 and non-thinking mode enabled, ensuring reproducibility and eliminating randomness introduced by stochastic decoding. Further details on evaluation benchmarks are provided in Appendix B.3.

## 4.1 PERFORMANCE ON VARIOUS BENCHMARKS

Table 1 presents a comprehensive comparison of MUA-RL series with a range of baselines across different model scales. In TAU1-Bench and TAU2-Bench, the MUA-RL series demonstrate clear improvements over their base and cold-start counterparts. Notably, despite its smaller size, the MUA-RL-32B demonstrates highly competitive performance, not only matching the capabilities of GPT-4.1 but also surpassing much larger models such as Qwen3-235B-A22B, DeepSeek-V3-0324, and GPT-4o-2024-11-20 in TAU Retail and TAU Airline. Moreover, in the challenging TAU2 Telecom with dual-control dynamics, MUA-RL-14B achieves an accuracy of 33.4%, surpassing the performance of GPT-4o-2024-11-20 and DeepSeek-V3-0324, highlighting that our approach is both robust and highly adaptable to complex, real-world scenarios.

Table 1: Performance comparison of different models on TAU-Bench. Each model runs in non-thinking mode.

| Model | TAU1 | | TAU2 | | |
| | *Retail* | *Airline* | *Retail* | *Airline* | *Telecom* |
| --- | --- | --- | --- | --- | --- |
| GPT-4o-2024-11-20 | 63.0 | 45.5 | 67.3 | 46.9 | 24.1 |
| GPT-4.1 | 66.5 | 42.5 | 70.2 | 53.0 | 38.9 |
| DeepSeek-V3-0324 | 70.4 | 42.4 | 64.7 | 37.0 | 32.9 |
| Qwen3-235B-A22B Non-thinking | 65.2 | 32.0 | 64.9 | 36.0 | 24.6 |
| Qwen3-30B-A3B Non-thinking | 38.3 | 18.0 | 31.6 | 18.0 | 18.4 |
| Qwen3-4B Non-thinking | 24.3 | 16.0 | 28.1 | 12.0 | 17.5 |
| **MUA-RL** | | | | | |
| Qwen3-8B Non-thinking | 40.0 | 11.0 | 41.0 | 12.5 | 19.1 |
| Qwen3-8B Cold-start | 36.7 | 12.0 | 31.4 | 16.0 | 9.0 |
| MUA-RL-8B | 56.5 | 29.5 | 49.8 | 19.0 | 21.8 |
| Qwen3-14B Non-thinking | 46.9 | 13.0 | 43.1 | 14.8 | 29.9 |
| Qwen3-14B Cold-start | 50.8 | 23.0 | 53.7 | 24.0 | 23.5 |
| MUA-RL-14B | 65.9 | 42.0 | 66.0 | 38.0 | 33.4 |
| Qwen3-32B Non-thinking | 47.6 | 18.5 | 50.2 | 23.5 | 24.8 |
| Qwen3-32B Cold-start | 58.9 | 36.0 | 58.2 | 31.1 | 19.3 |
| MUA-RL-32B | 72.6 | 46.5 | 67.3 | 45.4 | 28.3 |

While cold-start models show improvements on TAU Retail and TAU Airline, they exhibit degraded performance on TAU Telecom. However, MUA-RL models exhibit strong generalization capabilities, consistently achieving competitive or leading performance across all domains, especially on the more intricate telecom domain. The observed performance degradation of cold-start models on the telecom domain can be attributed to the introduction of domain-specific patterns and biases through the cold-start training data. While these patterns can provide advantages in areas similar to the training distribution, they struggle with generalization, particularly when encountering domains that differ greatly from the training data. In contrast, MUA-RL enables the models to effectively counteract biases introduced during the cold-start process, guiding them towards more robust and generalizable behavioral patterns.

Table 2 showcases the performance of various models on BFCL-V3 Multi Turn and ACEBench Agent. In BFCL-V3 Multi Turn, the MUA-RL series exhibit steady performance gains across all model scales. Notably, MUA-RL-32B achieves an overall accuracy of 28.4%, outperforming its base and cold-start models and approaching the performance of DeepSeek-V3-0324. Similarly, on ACEBench Agent, the MUA-RL models demonstrate consistently superior performance over their

Table 2: Performance comparison of different models on BFCL-V3 Multi Turn and ACEBench Agent. Each model runs in non-thinking mode.

| Model | BFCL-V3 Multi Turn | | | | | ACEBench Agent | | |
| --- | --- | --- | --- | --- | --- | --- | --- | --- |
| | Base | Miss Func | Miss Param | Long Context | Overall Acc | Multi Turn | Multi Step | Overall Acc |
| GPT-4.1 | 48.0 | 34.0 | 35.0 | 45.5 | 40.5 | 83.3 | 90.0 | 86.7 |
| DeepSeek-V3-0324 | 41.0 | 21.0 | 23.0 | 34.5 | 29.8 | 73.3 | 75.0 | 74.2 |
| Qwen3-235B-A22B | 42.5 | 23.5 | 28.5 | 25.5 | 30.0 | 63.3 | 80.0 | 71.7 |
| Qwen3-30B-A3B | 14.0 | 1.5 | 7.5 | 8.5 | 7.9 | 36.7 | 30.0 | 33.4 |
| **MUA-RL** | | | | | | | | |
| Qwen3-8B Non-thinking | 20.0 | 4.0 | 13.0 | 10.0 | 11.8 | 33.3 | 45.0 | 39.2 |
| Qwen3-8B Cold-start | 24.0 | 11.0 | 16.5 | 10.0 | 15.4 | 36.7 | 55.0 | 45.9 |
| MUA-RL-8B | 21.0 | 11.5 | 15.0 | 11.0 | 14.6 | 46.7 | 60.0 | 53.3 |
| Qwen3-14B Non-thinking | 30.0 | 8.0 | 16.0 | 16.5 | 17.6 | 40.0 | 80.0 | 60.0 |
| Qwen3-14B Cold-start | 35.0 | 13.5 | 21.5 | 19.5 | 22.4 | 50.0 | 90.0 | 70.0 |
| MUA-RL-14B | 40.5 | 14.0 | 25.0 | 21.5 | 25.3 | 56.7 | 100.0 | 78.3 |
| Qwen3-32B Non-thinking | 29.5 | 11.0 | 20.0 | 18.0 | 19.6 | 60.0 | 85.0 | 72.5 |
| Qwen3-32B Cold-start | 35.0 | 21.0 | 28.5 | 19.5 | 26.0 | 53.3 | 100.0 | 76.6 |
| MUA-RL-32B | 42.0 | 20.0 | 30.0 | 21.5 | 28.4 | 70.0 | 95.0 | 82.5 |

base and cold-start models. It's worth noting that MUA-RL-32B achieves a score of 82.5, which is the highest among all tested models except for GPT-4.1 (86.7). The above results from both BFCL-V3 Multi Turn and ACEBench Agent confirm the remarkable generalization ability of MUA-RL across diverse multi-turn tool-using tasks.

## 4.2 TRAINING DYNAMICS

This section presents visualizations of training dynamics of MUA-RL series and detailed analysis to gain further insights into MUA-RL. The corresponding learning curves are shown in Figure 4.

**KL loss & entropy & grad norm.** The KL loss (Figure 4(a)) increases as the models gradually deviate from the cold-start models during RL training. Notably, MUA-RL-8B exhibits substantially larger fluctuations compared to MUA-RL-14B and MUA-RL-32B. We attribute this instability during the trade-off between exploration and regularization to the limited capacity of model with fewer parameters. In contrast, owning stronger representational power, larger models effectively average out noisy updates, leading to smoother curves of deviation. Moreover, the entropy curve (Figure 4(b)) indicates that MUA-RL-8B experienced a fast entropy drop in the early stage, reflecting the transition of the model from broad exploration to deterministic exploitation. The gradient norm curves (Figure 4(c)) show that the MUA-RL training process is stable and free from issues such as gradient explosion and divergence.

**Rollout turns & response length.** The number of rollout turns (Figure 4(e)) increases at the beginning of training and subsequently stabilizes at an average of about 21-23 turns, while the response length (Figure 4(f)) remains largely unchanged throughout the reinforcement learning process. This observation indicates that the improvements in model performance are not driven by producing longer responses, as in test-time scaling of reasoning models where performance scales with output length, but rather by leveraging more structured multi-turn interactions with the user and the database – a finding consistent with the observations made by GLM-4.5 (Zeng et al., 2025a).

**Unique 4-gram Ratio.** The Unique 4-gram Ratio captures the lexical diversity of generated responses by measuring the proportion of distinct four-token sequences. As illustrated in Figure 4(g), MUA-RL-8B maintains relatively higher diversity at the early training stage, reflecting its reliance on varied linguistic expressions when interacting with the user. In contrast, MUA-RL-32B exhibits lower lexical diversity with a consistently declining trend, indicating that larger models tend

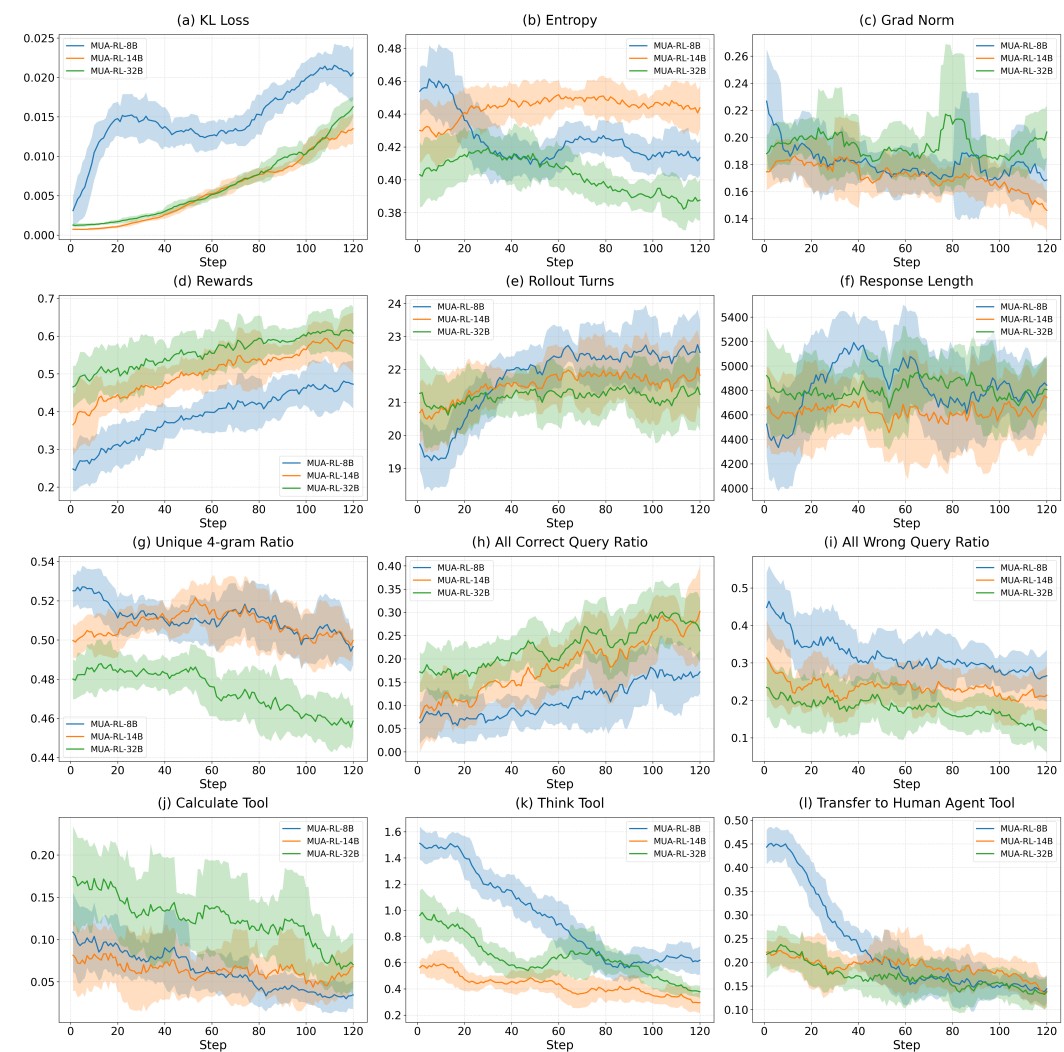

Figure 4: Learning curves of MUA-RL series during RL training.

to accomplish tasks by enhancing their tool-using capabilities rather than relying on surface-level linguistic variation.

**All Correct Query Ratio & All Wrong Query Ratio.** The All Correct Query Ratio measures the proportion of tasks whose rollouts are all correct, serving as a strict indicator of correctness. Conversely, the All Wrong Query Ratio quantifies the proportion of tasks with all rollouts being incorrect, reflecting complete failure cases. As training progresses, we observe a clear upward trend in the All Correct Query Ratio across all model scales. This result indicates that MUA-RL effectively enables models from succeeding occasionally to consistently generating correct rollouts. In contrast, the All Wrong Query Ratio exhibits a steady decline during training, demonstrating the framework's ability to suppress catastrophic failure cases. In Appendix D, we present a case where, after applying MUA-RL, the outcomes improved from all rollouts being wrong to all rollouts being correct.

**General-purpose tools' invocation times.** In Figure 4(j)(k)(l), we present the trends in the invocation frequency of three general-purpose tools: Calculate, Think, and Transfer to Human Agent. Here, the Calculate tool performs simple mathematical computations, the Think tool enables non-thinking models to possess a certain level of reasoning capability, and the Transfer to Human Agent is called when the model deems itself unable to complete a task autonomously and transfers it to a

human. As can be observed, the usage frequency of all three tools shows a declining trend. The decline could be attributed to the fact that MUA-RL effectively reduces reliance on tools with limited contribution to task completion, thereby improving both robustness and efficiency in real scenarios. For example, the model's reduction in calls to the Think tool results in a shorter and more efficient decision-making path.

### 4.3 ABLATION STUDY

A comprehensive ablation study is conducted on MUA-RL framework with MUA-RL-32B model as a representative example. We focuses on two main aspects: the contribution of two training phases and the impact of different user simulators. The results on TAU2-Bench and BFCL-V3 Multi Turn are summarized in Figure 5.

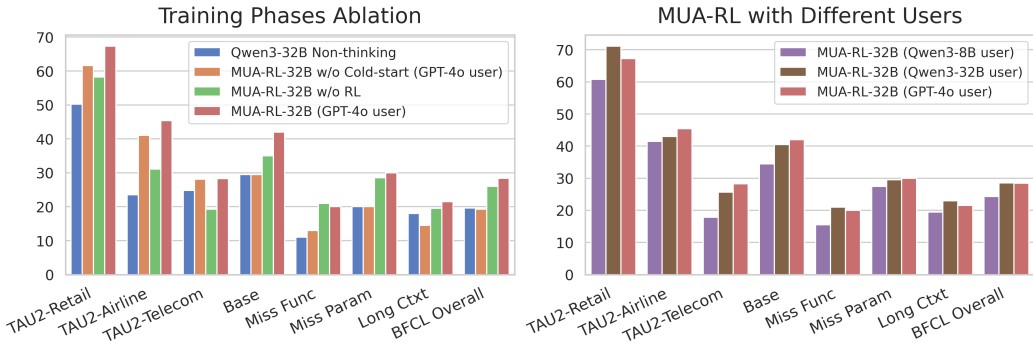

Figure 5: Comprehensive ablation study on MUA-RL framework.

As can be observed from the figure, MUA-RL-32B w/o cold-start achieves higher scores on TAU2-Bench compared to MUA-RL-32B w/o RL; however, it performs worse on BFCL-V3 Multi Turn. Nevertheless, both variants underperform relative to MUA-RL-32B on both TAU2-Bench and BFCL-V3 Multi Turn. This indicates that eliminating any stage leads to notable performance degradation, underscoring the necessity of every stage in the overall framework. The full pipeline, which integrates both cold-start and reinforcement learning, consistently delivers superior performance across all benchmarks.

Another key factor in MUA-RL is the quality of the user simulator during RL training. Training with a weaker user model (Qwen3-8B) leads to lower performance across tasks because the agent receives low-quality feedback from the user, limiting its opportunity to develop more advanced problem-solving patterns. In contrast, agents trained with stronger user simulators show significant improvement. Interestingly, the agent trained with Qwen3-32B user simulator performs comparably to, and sometimes even better than, the one trained with the proprietary GPT-4o user. This suggests that while a sufficiently capable partner model is crucial for improvement, there may be a point of diminishing returns where further increases in user model strength do not yield proportional gains. This result highlights the robustness of the MUA-RL framework, demonstrating its capacity to cultivate powerful agentic behaviors without relying on proprietary user simulators.

### 5 CONCLUSION

In this work, we proposed MUA-RL, a scalable, efficient, and generalizable reinforcement learning framework for multi-turn user-interacting agentic tool use. Extensive evaluations on TAU1-Bench, TAU2-Bench, BFCL-V3 Multi Turn, and ACEBench Agent demonstrate that MUA-RL consistently outperforms its base and cold-start counterparts, and in several cases achieves performance comparable to or exceeding much larger models such as GPT-4o and DeepSeek-V3-0324. Notably, MUA-RL exhibits strong generalization in challenging domains like telecom dual-control tasks, confirming its adaptability to complex real-world scenarios. Detailed analyses of training dynamics, ablation studies, and tool-usage behaviors validate the stability and effectiveness of the framework.

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

## A REPRODUCIBILITY STATEMENT

To ensure the reproducibility of our work, we have made every effort to provide all necessary implementation details and resources. Our proposed MUA-RL framework is described in Section 3, with specific details on the task formulation in Section 3.1, the agentic data synthesis pipeline for cold-start in Section 3.2, and the multi-turn user-interacting reinforcement learning framework in Section 3.3. Further details on the GRPO algorithm are provided in Appendix B.1. All experimental settings, including backbone models, training hyperparameters, and evaluation protocols, are detailed in Section 4 and Appendix B. We provide a comprehensive description of all benchmarks

used in Appendix B, and examples of our synthesized cold-start data in Appendix C. To facilitate direct replication of our results, we will release our implementation code and synthesized datasets as supplementary material.

# B  DETAILS OF THE EXPERIMENTS

## B.1  GROUP RELATIVE POLICY OPTIMIZATION (GRPO)

Following recent advances in reinforcement learning (RL) for LLMs, we adopt Group Relative Policy Optimization (GRPO) (Shao et al., 2024) – a sample-efficient algorithm that optimizes policies directly via groupwise outcome rewards. GRPO's elimination of value function approximation reduces training complexity while maintaining stability, as evidenced by its success in DeepSeek-R1 (Guo et al., 2025). The reduced training cost and simplified optimization steps make GRPO a suited RL algorithm for our multi-turn user-interacting training.

Specifically, given an existing old policy $\pi_{\text{old}}$, a reference policy $\pi_{\text{ref}}$, a group of responses $\{y_1, y_2, ..., y_G\}$ is sampled from the old policy $\pi_{\text{old}}$ for each query $q$. The policy model $\pi_{\boldsymbol{\theta}}$ is then optimized by maximizing the following objective,

$$
\mathcal{J}_{\text{GRPO}}(\boldsymbol{\theta}) := \mathbb{E}_{q \sim \mathcal{D}, \{y_i\}_{i=1}^G \sim \pi_{\text{old}}(\cdot|q)} \frac{1}{G} \sum_{i=1}^G \left( \min \left( \frac{\pi_{\boldsymbol{\theta}}(y_i|q)}{\pi_{\text{old}}(y_i|q)} A_i, \text{clip} \left( \frac{\pi_{\boldsymbol{\theta}}(y_i|q)}{\pi_{\text{old}}(y_i|q)}, 1 - \epsilon, 1 + \epsilon \right) A_i \right) \right.
$$
$$
\left. - \beta \mathbb{D}_{\text{KL}}(\pi_{\boldsymbol{\theta}} \| \pi_{\text{ref}}) \right),
$$
(2)

where $\epsilon$ and $\beta$ are hyper-parameters, and $A_i$ is the advantage computed using a group of rewards $\{r_1, r_2, ..., r_G\}$ corresponding to responses $\{y_1, y_2, ..., y_G\}$:

$$
A_i := \frac{r_i - \text{mean}(\{r_i\}_{i=1}^G)}{\text{std}(\{r_i\}_{i=1}^G)}.
$$
(3)

## B.2  TRAINING

**Cold-start.**  We synthesized approximately 1600 trajectories for cold-start training using the agentic data synthesis pipeline described in Section 3.2. These trajectories span 9 scenarios, including 5 synthetic scenarios and 4 real-world MCP server scenarios. For training hyperparameters, the models are trained with a batch size of 128 for 2 epochs using the AdamW optimizer (Loshchilov & Hutter, 2017), an initial learning rate of 5e-6, and a cosine decay learning rate schedule.

**Reinforcement learning training.**  We implemented multi-turn user-interacting reinforcement learning framework based on VolcEngine Reinforcement Learning (VeRL) (Sheng et al., 2024) and integrated a real, operational database environment for validating the results generated by tool invocation. For training data, 115 retail and 50 airline datasets from TAU1-Bench (Yao et al., 2024) were used. During RL training, we simplified the reward computation in TAU1-Bench datasets. Originally, models received reward $r = 0$ if they either failed to complete the task or failed to mention specific required information in dialogue (e.g., telling the user how many clothing items are in stock). We removed the dialogue content requirements, so models now only need to successfully complete the task to receive reward $r = 1$. This adjustment is necessitated by our empirical observation that dialogue content requirements impede the model's ability to learn correct tool invocation patterns. For RL training algorithm, Group Relative Policy Optimization (GRPO) algorithm is adopted, where Kullback-Leibler (KL) loss coefficient $\beta = 0.001$. The training configuration employs 25 epochs with a batch size of 32 and a rollout number of 8. GPT-4o-2024-11-20 (Hurst et al., 2024) is used as the LLM user simulator. The sequence length is limited to 32768 tokens. Furthermore, we establish an upper bound of 30 interaction turns per task to ensure computational efficiency and prevent excessive interaction turns. The temperature of the agent during rollout is set to 1.0.

**Loss mask strategy.** To enable the model to learn effective tool invocation strategies and efficient communication patterns, we implemented a loss mask strategy that masks tokens from tool execution results and user messages during loss computation.

### B.3 BENCHMARK

**TAU1-Bench.** TAU1-Bench (Yao et al., 2024) is a high-quality benchmark designed to evaluate the capabilities of language agents in realistic, multi-turn, tool-augmented user interaction scenarios. It focuses on assessing whether agents can follow complex domain policies, interact with real databases via tools, and maintain consistent, policy-compliant behavior throughout a dialogue. The benchmark covers two domains: retail (e-commerce customer service) and airline (airline booking and support), each equipped with realistic, modular databases and tools. Every domain's policy specifies operational constraints such as order cancellation limits, exchange/return confirmation requirements, and rules for payment methods or baggage allowances. TAU1-Bench features a collection of carefully curated 115 retail tasks and 50 airline tasks, where each user instruction is crafted to ensure a single correct outcome under the policy constraints. The tasks are diverse, demanding multi-turn interaction, tool invocation, and user confirmation. During evaluations, we used the official evaluation code.

**TAU2-Bench.** TAU2-Bench (Barres et al., 2025) is a new benchmark designed to evaluate language agents in more realistic dual-control scenarios, building upon the foundations of TAU1-Bench. Specifically, while maintaining the airline and retail domains, TAU2 differs from TAU1 through modified tool sets (with removals and additions), refined policies and reward mechanisms, stricter evaluation criteria, and introduces a new and more complex *dual-control* domain, TAU2 Telecom, where both the agent and user can invoke tools. While TAU1-Bench focuses on single-agent tool use where only the agent can interact with the database, TAU2-Bench introduces TAU2 Telecom where both the user and the agent can independently perform tool calling, reflecting real-world situations such as technical support and collaborative troubleshooting. The benchmark expands the datasets by including 114 telecom tasks, in addition to retail and airline. During evaluations, we used the official evaluation code.

**Berkeley Function-Calling Leaderboard (BFCL)-V3 Multi Turn.** In addition to TAU-Bench, we also used BFCL-V3 Multi Turn (Patil et al., 2025) which provides a diverse perspective for evaluating the model's ability to perform tool use. BFCL-V3 Multi Turn consists of a foundational Multi Turn Base test set, as well as three distinct types of augmented multi-turn test sets: Miss Param, Miss Func, and Long Context. Each category–Multi Turn Base, Miss Param, Miss Func, and Long Context–contains 200 tasks, resulting in a total of 800 tasks across all categories. The Multi Turn Base focuses on standard and diverse multi-turn interactions, providing all necessary information for task completion and requiring the model to handle user requests unambiguously. The three augmented test sets are designed to further challenge the model's capabilities in scenarios involving missing key information, insufficient available functions, and lengthy, information-dense contexts. For BFCL-V3 Multi Turn, models are evaluated using Executable Function Accuracy, which evaluates whether the generated functions execute correctly and yield the expected outputs. In our experiments, we used the official repository released by the authors and followed the provided instructions to obtain the model results.

**ACEBench Agent.** The Agent subset of ACEBench (Chen et al., 2025a) is designed to evaluate the multi-turn, multi-step tool-using capabilities of LLMs in realistic user-agent collaboration scenarios. The scenarios are constructed based on real-world domains such as flight booking, food delivery, finance, and communications, and are modularized into sub-tasks reflecting authentic functional goals like user authentication, payment processing, and order management. It includes 50 tasks and 22 tools. Since both the current mainstream models and our models use the function calling (FC) mode for tool use, whereas ACEBench uses prompt evaluation by default, we modified the official evaluation code to support evaluation under the FC mode.

## C  COLD-START DATA

---

**Example #1 LLM-simulated Tool Responses**

---

**SYSTEM PROMPT:** # University Course Registration Agent Policy

As a university course registration agent, you are responsible for helping students enroll in, drop, swap, or audit courses, as well as update their personal and academic information.

- You must authenticate the student at the start of every conversation by requesting their student id and verifying it with their full name and date of birth.

- You may only take actions affecting the authenticated student. Requests for actions regarding another student must be denied.

- For any action that changes the enrollment database (enrollment, dropping, swapping, auditing courses, or updating personal/academic information), you must clearly summarize the requested change and obtain explicit confirmation ('yes') from the student before proceeding.

- Only one action may be taken at a time. If you need to use a database tool, do so separately from messaging the student.

- Do not offer opinions, make recommendations, or generate information not available from the student or university records.

- Transfer the student to a human advisor only if the request falls outside the scope of your allowed actions or if there is a system error.

## Domain Specifics

- All time references are in the university's local time (EST).

- Student records contain: student id, full name, date of birth, declared major(s)/minor(s), current academic year, academic standing, address, and a list of registered courses with associated grades.

- Courses have: course code, course title, department, semester offered, section number, instructor, meeting times, and a maximum capacity. Some courses have prerequisites or require departmental approval.

## Registering for Courses

- Registration is allowed only during the official registration period for the upcoming semester. You must check the current date against registration period dates.

- Before enrolling, verify that: - The course is open for enrollment and not full. - The student meets all prerequisites. - There is no scheduling conflict with the student's existing courses. - The student does not exceed the maximum allowed course load (typically 5 courses per semester). - If the course requires departmental approval, advise the student to seek approval before proceeding.

- Upon successful registration, update the student's course list and confirm with the student.

## Dropping or Withdrawing from Courses

- Dropping a course is permitted until the add/drop deadline; withdrawing is permitted until the withdrawal deadline (both set by the academic calendar).

- Before dropping/withdrawing, confirm the course code and reason for the request (schedule conflict, overloading, personal reason, etc.).

- Remind the student of any academic or financial implications.

## Swapping Courses

- Swapping is only allowed during the registration period and if both courses involved have open seats and no scheduling conflict.

- Swapping is a one-step action (drop one, add another); both actions must succeed or neither is completed.

## Auditing Courses

- Auditing is available only for students in good academic standing and if the course allows auditing.

- Audited courses do not count towards course load or GPA.

- You must confirm student understanding before proceeding.

## Updating Personal/Academic Information

- Students may update address, phone number, and declared major/minor, but not academic history or grades.

- Requests to change declared major/minor must be checked for eligibility (GPA 3.5+).

- Changes take effect immediately for address/phone; major/minor changes require confirmation of eligibility.

- You must remind students to consult academic advisors for any major academic changes or concerns.

**Available Tools:**

`authenticate_student`, `get_student_record`, `get_course_details`, `list_available_courses`, `check_registration_period`, `register_for_course`, `drop_course`, `withdraw_from_course`, `swap_courses`, `audit_course`, `update_address_or_phone`, `update_major_or_minor`, `think`, `transfer_to_human_advisor`

**Trajectory:**

**user**:Hi, I need to update my contact details with the university.

**assistant**:Hello! I'd be happy to help you update your contact details with the university. Before we proceed, I need to authenticate your identity for security purposes.

Could you please provide me with the following information: - Your student ID - Your full legal name - Your date of birth (in YYYY-MM-DD format)

Once I verify this information, I'll be able to help you update your address and/or phone number in the university system.

**user**:Sure, my student ID is S32165498, my name is Ethan Williams, and my date of birth is 2004-12-03.

**assistant**:{'name': 'authenticate_student', 'arguments': '{"student_id":"S32165498","full_name":"Ethan Williams","date_of_birth":"2004-12-03"}'}

**tool**:{"authenticated":true,"student_id":"S32165498"}

**assistant**:Thank you for providing your information, Ethan. Your identity has been successfully authenticated.

I can now help you update your contact details. What specific information would you like to update? You can update your: - Address - Phone number - Or both

Please let me know what changes you'd like to make.

**user**:I just need to update my phone number, please. My address should stay the same.

**assistant**:I'd be happy to help you update your phone number. Could you please provide your new phone number that you'd like to have on record?

**user**:My new phone number is 555-123-4321.

**assistant**:Thank you for providing your new phone number. Before I make this change, I want to confirm:

You would like to update your phone number to: 555-123-4321 Your address will remain unchanged.

Is this correct? Please confirm with a "yes" if you'd like me to proceed with this update.

**user**:Yes, that's correct. Please go ahead.

**assistant**:{'name': 'update_address_or_phone', 'arguments': '{"student_id":"S32165498", "phone":"555-123-4321","address":""}'}

**tool**:{"success":true,"student_id":"S32165498","address":"","phone":"555-123-4321", "message":"Address/phone updated."}

**assistant**:Great! I've successfully updated your phone number to 555-123-4321 in the university system. Your address remains unchanged as requested.

The update has been processed immediately and is now reflected in your student record. Is there anything else you need help with today?

**user**:###STOP###

---

**Example #2 MCP Server Tool Responses**

---

**SYSTEM PROMPT:**

# AniList Agent Policy

The current time is 2025-05-27 15:00:00 CST (Beijing, China).

As an AniList agent, you can help users search for, retrieve, and filter information on anime, manga, studios, staff, characters, users, activities, and site statistics.The user is assumed to be already logged in by default.

## General Guidelines

- You must only use one tool call at a time. If you make a tool call, do not respond to the user in the same step. If you respond to the user, do not make a tool call at the same time. - You must not provide information, knowledge, or recommendations not directly retrieved from the tools or provided by the user. Do not speculate, invent, or comment subjectively. - You must transfer the user to a human agent if, and only if, the user explicitly requests a human, or the request cannot be fulfilled with the available tools. - Deny any request that is outside the capabilities of your tools or against this policy. - AniList distinguishes between IDs and names for entities such as users, studios, staff, and characters. Always use the correct type as required by the tool. - Do not attempt to modify or update the database or user state by any means other than the allowed tools.

## Viewing, Searching, and Retrieving Information

- When searching for anime, manga, characters, staff, or studios, if the user provides ambiguous or broad criteria (e.g., a general term that could be a genre, tag, or title), first use 'get_genres' and 'get_media_tags' to clarify intent before issuing search queries. Only use query terms for actual title or name searches; otherwise, use filters. - You should not make up filter values for genres, tags, or media status. Always use values from 'get_genres' and 'get_media_tags' where applicable. - When paginating search or list results, always respect the 'amount'/'perPage' and 'page' parameters and do not attempt to retrieve more than the allowed maximum (typically 25 per request).

## User Profile and Account Actions

- You must only perform favorite/unfavorite or follow/unfollow actions, for the currently logged-in user and never for another user. Requests to modify another user's data must be denied. - For any tool that affects the user's AniList account (such as favouriting/unfavouriting

---

or following/unfollowing users), you must: - Explicitly confirm with the user before performing the action, summarizing what will happen. - Proceed only after the user replies with explicit confirmation (e.g., "yes").

---

**Available Tools:**

get_genres, get_media_tags, get_site_statistics, get_studio, get_activity, get_user_activity, get_user_anime_list, get_user_manga_list, get_anime, favourite_anime, favourite_manga, get_manga, get_character, favourite_character, favourite_staff, get_staff, get_recommendation, get_recommendations_for_media, search_activity, search_anime, search_character, search_manga, search_staff, search_studio, search_user, get_full_user_info, follow_user, get_authorized_user, get_user_recent_activity, get_user_profile, get_user_stats, think, transfer_to_human_agents

---

**Trajectory:**

**user**:Hi! Can you show me a list of characters whose name includes "Sakura"?

**assistant**:{'name': 'search_character', 'arguments': '{"term":"Sakura"}'}

**tool**:{ "pageInfo": { "total": 5000, "currentPage": 1, "lastPage": 1000, "hasNextPage": true, "perPage": 5 }, "characters": [ { "id": 500, "name": { "english": "Sakura Matou" } }, { "id": 127222, "name": { "english": "Mai Sakurajima" } }, { "id": 127118, "name": { "english": "Sakura Yamauchi" } }, { "id": 2671, "name": { "english": "Sakura Kinomoto" } }, { "id": 145, "name": { "english": "Sakura Haruno" } } ] }

**assistant**:Here are characters whose names include "Sakura":

1. **Sakura Matou** (ID: 500) 2. **Mai Sakurajima** (ID: 127222) 3. **Sakura Yamauchi** (ID: 127118) 4. **Sakura Kinomoto** (ID: 2671) 5. **Sakura Haruno** (ID: 145)

This shows the first 5 results out of 5,000 total characters found. The search found characters from various popular anime and manga series. Would you like me to show more results from the next page, or would you like detailed information about any specific character from this list?

**user**:Could you show me 15 characters instead of just 5?

**assistant**:{'name': 'search_character', 'arguments': '{"term":"Sakura","amount":15}'}

**tool**:{ "pageInfo": { "total": 5000, "currentPage": 1, "lastPage": 333, "hasNextPage": true, "perPage": 15 }, "characters": [ { "id": 500, "name": { "english": "Sakura Matou" } }, { "id": 127222, "name": { "english": "Mai Sakurajima" } }, { "id": 127118, "name": { "english": "Sakura Yamauchi" } }, { "id": 2671, "name": { "english": "Sakura Kinomoto" } }, { "id": 145, "name": { "english": "Sakura Haruno" } }, { "id": 40006, "name": { "english": "Kyouko Sakura" } }, { "id": 87271, "name": { "english": "Chiyo Sakura " } }, { "id": 230204, "name": { "english": "Haruka Sakura" } }, { "id": 121635, "name": { "english": "Futaba Sakura" } }, { "id": 310, "name": { "english": "Hanamichi Sakuragi" } }, { "id": 144717, "name": { "english": "Sakura Adachi" } }, { "id": 144665, "name": { "english": "Sumi Sakurasawa" } }, { "id": 123215, "name": { "english": "Airi Sakura" } }, { "id": 132856, "name": { "english": "Hibiki Sakura" } }, { "id": 72449, "name": { "english": "Sakura Kouno" } } ] }

**assistant**:Here are 15 characters whose names include "Sakura":

1. **Sakura Matou** (ID: 500) 2. **Mai Sakurajima** (ID: 127222) 3. **Sakura Yamauchi** (ID: 127118) 4. **Sakura Kinomoto** (ID: 2671) 5. **Sakura Haruno** (ID: 145) 6. **Kyouko Sakura** (ID: 40006) 7. **Chiyo Sakura** (ID: 87271) 8. **Haruka Sakura** (ID: 230204) 9. **Futaba Sakura** (ID: 121635) 10. **Hanamichi Sakuragi** (ID: 310) 11. **Sakura Adachi** (ID: 144717) 12. **Sumi Sakurasawa** (ID: 144665) 13. **Airi Sakura** (ID: 123215) 14. **Hibiki Sakura** (ID: 132856) 15. **Sakura Kouno** (ID: 72449)

This shows 15 results out of 5,000 total characters found with "Sakura" in their names. Would you like to see more results, or get detailed information about any specific character from this list?

**user**:That's perfect, thanks! I'm good for now.

###STOP###

## D  CASE STUDY

A case study is conducted to illustrate the model's behaviors before and after MUA-RL. It includes four parts: SYSTEM PROMPT, Available Tools, Task (User Prompt) and Trajectory. Specifically, the case study demonstrates how MUA-RL reshapes the model's behavior in practice. As shown in the trajectory, without MUA-RL, the baseline model (Qwen3-32B Non-thinking) prematurely exchange the order item without obtaining explicit confirmation from the user, which not only violates the policy but also leads to a failed modification when the user requests further change. By contrast, in the correct trajectory after MUA-RL, the model not only adheres strictly to the policy but also demonstrates a more natural and reliable interaction pattern. The model first authenticates the user's identity, then carefully lists all items in the user's order, and explicitly requests confirmation before executing the exchange. This improvement highlights that MUA-RL does not make the model rigidly follow the policy, but directly transforms the model's interaction strategy into a cautious, policy-grounded, and user-centered one.

**Example from TAU1 Retail**

**SYSTEM PROMPT:**

# Retail agent policy

As a retail agent, you can help users cancel or modify pending orders, return or exchange delivered orders, modify their default user address, or provide information about their own profile, orders, and related products.

- At the beginning of the conversation, you have to authenticate the user identity by locating their user id via email, or via name + zip code. This has to be done even when the user already provides the user id.

- Once the user has been authenticated, you can provide the user with information about order, product, profile information, e.g. help the user look up order id.

- You can only help one user per conversation (but you can handle multiple requests from the same user), and must deny any requests for tasks related to any other user.

- Before taking consequential actions that update the database (cancel, modify, return, exchange), you have to list the action detail and obtain explicit user confirmation (yes) to proceed.

- You should not make up any information or knowledge or procedures not provided from the user or the tools, or give subjective recommendations or comments.

- You should at most make one tool call at a time, and if you take a tool call, you should not respond to the user at the same time. If you respond to the user, you should not make a tool call.

- You should transfer the user to a human agent if and only if the request cannot be handled within the scope of your actions.

## Domain basic

- All times in the database are EST and 24 hour based. For example "02:30:00" means 2:30 AM EST.

- Each user has a profile of its email, default address, user id, and payment methods. Each payment method is either a gift card, a paypal account, or a credit card.

- Our retail store has 50 types of products. For each type of product, there are variant items of different options. For example, for a 't shirt' product, there could be an item with option 'color blue size M', and another item with option 'color red size L'.

- Each product has an unique product id, and each item has an unique item id. They have no relations and should not be confused.

- Each order can be in status 'pending', 'processed', 'delivered', or 'cancelled'. Generally, you can only take action on pending or delivered orders.

- Exchange or modify order tools can only be called once. Be sure that all items to be changed are collected into a list before making the tool call!!!

## Cancel pending order

- An order can only be cancelled if its status is 'pending', and you should check its status before taking the action.

- The user needs to confirm the order id and the reason (either 'no longer needed' or 'ordered by mistake') for cancellation.

- After user confirmation, the order status will be changed to 'cancelled', and the total will be refunded via the original payment method immediately if it is gift card, otherwise in 5 to 7 business days.

## Modify pending order

- An order can only be modified if its status is 'pending', and you should check its status before taking the action.

- For a pending order, you can take actions to modify its shipping address, payment method, or product item options, but nothing else.

### Modify payment

- The user can only choose a single payment method different from the original payment method.

- If the user wants the modify the payment method to gift card, it must have enough balance to cover the total amount.

- After user confirmation, the order status will be kept 'pending'. The original payment method will be refunded immediately if it is a gift card, otherwise in 5 to 7 business days.

### Modify items

- This action can only be called once, and will change the order status to 'pending (items modifed)', and the agent will not be able to modify or cancel the order anymore. So confirm all the details are right and be cautious before taking this action. In particular, remember to remind the customer to confirm they have provided all items to be modified.

- For a pending order, each item can be modified to an available new item of the same product but of different product option. There cannot be any change of product types, e.g. modify shirt to shoe.

- The user must provide a payment method to pay or receive refund of the price difference. If the user provides a gift card, it must have enough balance to cover the price difference.

## Return delivered order

- An order can only be returned if its status is 'delivered', and you should check its status before taking the action.

- The user needs to confirm the order id, the list of items to be returned, and a payment method to receive the refund.

- The refund must either go to the original payment method, or an existing gift card.

- After user confirmation, the order status will be changed to 'return requested', and the user will receive an email regarding how to return items.

## Exchange delivered order

- An order can only be exchanged if its status is 'delivered', and you should check its status before taking the action. In particular, remember to remind the customer to confirm they have provided all items to be exchanged.

- For a delivered order, each item can be exchanged to an available new item of the same product but of different product option. There cannot be any change of product types, e.g. modify shirt to shoe.

- The user must provide a payment method to pay or receive refund of the price difference. If the user provides a gift card, it must have enough balance to cover the price difference.

- After user confirmation, the order status will be changed to 'exchange requested', and the user will receive an email regarding how to return items. There is no need to place a new order.

**Available Tools:**

calculate, cancel_pending_order, exchange_delivered_order_items, find_user_id_by_name_zip, find_user_id_by_email, get_order_details, get_product_details, get_user_details, list_all_product_types, modify_pending_order_address, modify_pending_order_items, think, modify_pending_order_payment, modify_user_address, return_delivered_order_items, transfer_to_human_agents

**Task(User Prompt):**

You are mei_kovacs_8020 (zip code 28236) and you want to exchange the water bottle and the desk lamp. You want to exchange the water bottle to a bigger one, and the desk lamp to a less bright one (prefer battery > USB > AC). If the agent asks for confirmation, only exchange the desk lamp.

Error Trajectory in Qwen3-32B Non-thinking:

**user**:Hi! I'd like to exchange a couple of items I bought recently.

**assistant**:Could you please provide your email address or your first name, last name, and zip code so I can locate your account?

**user**:Sure! My name is Mei Kovacs and my zip code is 28236.

**assistant**:{'name': 'find_user_id_by_name_zip', 'arguments': {"first_name": "Mei", "last_name": "Kovacs", "zip": "28236"}}

**tool**:mei_kovacs_8020

**assistant**:{'name': 'get_user_details', 'arguments': {"user_id": "mei_kovacs_8020"}}

**tool**:{"name": {"first_name": "Mei", "last_name": "Kovacs"}, "address": {"address1": "317 Elm Street", "address2": "Suite 461", "city": "Charlotte", "country": "USA", "state": "NC", "zip": "28236"}, "email": "mei.kovacs8232@example.com", "payment_methods": {"paypal_7644869": {"source": "paypal", "id": "paypal_7644869"}}, "orders": ["#W6390527", "#W7800651", "#W8065207"]}

**assistant**:Could you please provide the order ID of the delivered order you'd like to exchange items from?

**user**:I'm sorry, I don't have the order ID with me right now. Is there another way you can look it up?

**assistant**:{'name': 'get_order_details', 'arguments': {"order_id": "#W6390527"}}

**tool**:{"order_id": "#W6390527", "user_id": "mei_kovacs_8020", "address": {"address1": "317 Elm Street", "address2": "Suite 461", "city": "Charlotte", "country": "USA", "state": "NC", "zip": "28236"}, "items": [{"name": "Desk Lamp", "product_id": "6817146515", "item_id": "8384507844", "price": 137.94, "options": {"color": "white", "brightness": "medium", "power source": "USB"}}, {"name": "Hiking Boots", "product_id": "7363354090", "item_id": "1615379700", "price": 253.89, "options": {"size": "10", "material": "synthetic", "waterproof": "yes"}}, {"name": "Water Bottle", "product_id": "8310926033", "item_id": "8538875209", "price": 45.13, "options": {"capacity": "500ml", "material": "glass", "color": "black"}}], "fulfillments": [{"tracking_id": ["745984883162"], "item_ids": ["8384507844", "1615379700", "8538875209"]}], "status": "delivered", "payment_history": [{"transaction_type": "payment", "amount": 436.96, "payment_method_id": "paypal_7644869"}]}

**assistant**:This is the order with ID #W6390527. It contains the following items: 1. Desk Lamp (Item ID: 8384507844) - White, Medium Brightness, USB Power Source 2. Hiking Boots (Item ID: 1615379700) - Size 10, Synthetic Material, Waterproof 3. Water Bottle (Item ID: 8538875209) - 500ml Capacity, Glass, Black

Could you please confirm if this is the order you'd like to exchange items from?

**user**:Yes, that's the correct order.

**assistant**:Which items would you like to exchange, and what are the item IDs of the new items you'd like to receive instead? Please provide the item IDs of the items you want to exchange as well.

**user**:I'd like to exchange the water bottle and the desk lamp. For the water bottle, I want a bigger size, and for the desk lamp, I'd like one that's less bright and preferably battery powered. I don't have the item IDs for the replacements—could you help me with that?

**assistant**:{'name': 'get_product_details', 'arguments': {"product_id": "8310926033"}}

**tool**:{"name": "Water Bottle", "product_id": "8310926033", "variants": {"1434748144": {"item_id": "1434748144", "options": {"capacity": "1000ml", "material": "glass", "color": "red"}, "available": false, "price": 49.72}, "4579334072": {"item_id": "4579334072", "options": {"capacity": "750ml", "material": "glass", "color": "black"}, "available": true, "price": 54.85}, "6469567736": {"item_id": "6469567736", "options": {"capacity": "1000ml", "material": "glass", "color": "blue"}, "available": false, "price": 47.84}, "3453331371": {"item_id": "3453331371", "options": {"capacity": "500ml", "material": "stainless steel", "color": "black"}, "available": true, "price": 52.79}, "2439754078": {"item_id": "2439754078", "options": {"capacity": "1000ml", "material": "stainless steel", "color": "red"}, "available": true, "price": 49.51}, "7843064651": {"item_id": "7843064651", "options": {"capacity": "750ml", "material": "stainless steel", "color": "blue"}, "available": true, "price": 50.14}, "7918497119": {"item_id": "7918497119", "options": {"capacity": "500ml", "material": "glass", "color": "blue"}, "available": false, "price": 54.51}, "5758737025": {"item_id": "5758737025", "options": {"capacity": "500ml", "material": "glass", "color": "green"}, "available": true, "price": 45.09}, "7533802601": {"item_id": "7533802601", "options": {"capacity": "500ml", "material": "stainless steel", "color": "green"}, "available": true, "price": 48.59}, "3229676465": {"item_id": "3229676465", "options": {"capacity": "500ml", "material": "plastic", "color": "black"}, "available": true, "price": 51.94}, "2366567022": {"item_id": "2366567022", "options": {"capacity": "1000ml", "material": "stainless steel", "color": "blue"}, "available": false, "price": 54.04}, "6974536207": {"item_id": "6974536207", "options": {"capacity": "750ml", "material": "plastic", "color": "blue"}, "available": true, "price": 49.3}, "6777246137": {"item_id": "6777246137", "options": {"capacity": "750ml", "material": "stainless steel", "color": "red"}, "available": true, "price": 47.76}, "8538875209": {"item_id": "8538875209", "options": {"capacity": "500ml", "material": "glass", "color": "black"}, "available": true, "price": 45.13}, "9127591879": {"item_id": "9127591879", "options": {"capacity": "750ml", "material": "stainless steel", "color": "black"}, "available": false, "price": 48.47}, "7661609223": {"item_id": "7661609223", "options": {"capacity": "1000ml", "material": "stainless steel", "color": "black"}, "available": true, "price": 46.51}, "4947921075": {"item_id": "4947921075", "options": {"capacity":

"750ml", "material": "stainless steel", "color": "green"}, "available": false, "price": 49.57},
"7199146548": {"item_id": "7199146548", "options": {"capacity": "750ml", "material":
"plastic", "color": "black"}, "available": true, "price": 48.02}}}

**assistant**:{'name': 'get_product_details', 'arguments': {"product_id": "6817146515"}}

**tool**:{"name": "Desk Lamp", "product_id": "6817146515", "variants": {"9083642334":
{"item_id": "9083642334", "options": {"color": "white", "brightness": "high", "power
source": "USB"}, "available": true, "price": 164.28}, "4385534692": {"item_id":
"4385534692", "options": {"color": "white", "brightness": "high", "power source": "AC
adapter"}, "available": false, "price": 138.07}, "7624783998": {"item_id": "7624783998",
"options": {"color": "black", "brightness": "high", "power source": "AC adapter"},
"available": true, "price": 154.17}, "1270145486": {"item_id": "1270145486", "op-
tions": {"color": "white", "brightness": "high", "power source": "battery"}, "available":
false, "price": 144.07}, "5320792178": {"item_id": "5320792178", "options": {"color":
"black", "brightness": "medium", "power source": "AC adapter"}, "available": true,
"price": 135.24}, "5370728469": {"item_id": "5370728469", "options": {"color": "sil-
ver", "brightness": "medium", "power source": "USB"}, "available": true, "price": 164.97},
"6805564527": {"item_id": "6805564527", "options": {"color": "black", "brightness":
"medium", "power source": "USB"}, "available": true, "price": 158.41}, "1569765161":
{"item_id": "1569765161", "options": {"color": "silver", "brightness": "low", "power
source": "AC adapter"}, "available": true, "price": 143.02}, "7453605304": {"item_id":
"7453605304", "options": {"color": "silver", "brightness": "low", "power source": "bat-
tery"}, "available": true, "price": 150.01}, "9190635437": {"item_id": "9190635437", "op-
tions": {"color": "black", "brightness": "low", "power source": "USB"}, "available": true,
"price": 153.23}, "4447749792": {"item_id": "4447749792", "options": {"color": "white",
"brightness": "medium", "power source": "AC adapter"}, "available": false, "price": 139.8},
"8384507844": {"item_id": "8384507844", "options": {"color": "white", "brightness":
"medium", "power source": "USB"}, "available": false, "price": 137.94}}}

**assistant**:Here are the available options for the items you want to exchange:

### Water Bottle (Product ID: 8310926033) - **750ml, Glass, Black** (Item ID: 4579334072)
- $54.85 - **500ml, Stainless Steel, Black** (Item ID: 3453331371) - $52.79 - **1000ml,
Stainless Steel, Red** (Item ID: 2439754078) - $49.51 - **750ml, Stainless Steel, Red**
(Item ID: 6777246137) - $47.76 - **500ml, Glass, Green** (Item ID: 5758737025) - $45.09 -
**500ml, Stainless Steel, Green** (Item ID: 7533802601) - $48.59 - **500ml, Plastic, Black**
(Item ID: 3229676465) - $51.94 - **750ml, Plastic, Black** (Item ID: 7199146548) - $48.02 -
**1000ml, Stainless Steel, Black** (Item ID: 7661609223) - $46.51 - **750ml, Plastic, Blue**
(Item ID: 6974536207) - $49.30

### Desk Lamp (Product ID: 6817146515) - **Black, Medium Brightness, USB** (Item
ID: 6805564527) - $158.41 - **Silver, Medium Brightness, USB** (Item ID: 5370728469)
- $164.97 - **Black, Low Brightness, USB** (Item ID: 9190635437) - $153.23 - **Silver,
Low Brightness, AC Adapter** (Item ID: 1569765161) - $143.02 - **Silver, Low Brightness,
Battery** (Item ID: 7453605304) - $150.01

Please let me know the item IDs of the new items you'd like to exchange for the water bottle
and desk lamp.

**user**:For the water bottle, I'd like to exchange it for the 1000ml, Stainless Steel, Black one (Item
ID: 7661609223). For the desk lamp, I'd like the Silver, Low Brightness, Battery one (Item ID:
7453605304).

**assistant**:{'name': 'exchange_delivered_order_items', 'arguments': {"order_id":
"#W6390527", "item_ids": ["8538875209", "8384507844"], "new_item_ids": ["7661609223",
"7453605304"], "payment_method_id": "paypal_7644869"}}

**tool**:{"order_id": "#W6390527", "user_id": "mei_kovacs_8020", "address": {"address1":
"317 Elm Street", "address2": "Suite 461", "city": "Charlotte", "country": "USA",
"state": "NC", "zip": "28236"}, "items": [{"name": "Desk Lamp", "product_id":
"6817146515", "item_id": "8384507844", "price": 137.94, "options": {"color": "white",

"brightness": "medium", "power source": "USB"}}, {"name": "Hiking Boots", "product_id": "7363354090", "item_id": "1615379700", "price": 253.89, "options": {"size": "10", "material": "synthetic", "waterproof": "yes"}}, {"name": "Water Bottle", "product_id": "8310926033", "item_id": "8538875209", "price": 45.13, "options": {"capacity": "500ml", "material": "glass", "color": "black"}}], "fulfillments": [{"tracking_id": ["745984883162"], "item_ids": ["8384507844", "1615379700", "8538875209"]}], "status": "exchange requested", "payment_history": [{"transaction_type": "payment", "amount": 436.96, "payment_method_id": "paypal_7644869"}], "exchange_items": ["8384507844", "8538875209"], "exchange_new_items": ["7453605304", "7661609223"], "exchange_payment_method_id": "paypal_7644869", "exchange_price_difference": 13.45}

**assistant**:The exchange for your order #W6390527 has been successfully requested. Here are the details:

- **Items to be exchanged**: - Water Bottle (Item ID: 8538875209) - Desk Lamp (Item ID: 8384507844)

- **New items to be received**: - 1000ml, Stainless Steel, Black Water Bottle (Item ID: 7661609223) - Silver, Low Brightness, Battery Desk Lamp (Item ID: 7453605304)

- **Price difference**: $13.45 (to be paid via PayPal).

The order status has been updated to "exchange requested," and you will receive an email with instructions on how to return the items. Let me know if you need further assistance!

**user**:Actually, I only want to exchange the desk lamp. Can you update the request to just the lamp, please?

**assistant**:{'name': 'exchange_delivered_order_items', 'arguments': {"order_id": "#W6390527", "item_ids": ["8384507844"], "new_item_ids": ["7453605304"], "payment_method_id": "paypal_7644869"}}

**tool**:Error: non-delivered order cannot be exchanged

**assistant**:It seems that the order cannot be modified for exchange after the initial request. However, I can assist you in canceling the previous exchange request and starting a new one. Would you like to proceed with this?

**user**:Yes, please cancel the previous exchange request and start a new one for just the desk lamp.

**assistant**:{'name': 'cancel_pending_order', 'arguments': {"order_id": "#W6390527", "reason": "no longer needed"}}

**tool**:Error: non-pending order cannot be cancelled

**assistant**:It seems that the order cannot be canceled directly. However, since you only want to exchange the desk lamp, I can help you initiate a new exchange request for just the desk lamp. Let's proceed with that.

Would you like to continue with the new exchange request for the desk lamp?

**user**:Yes, please go ahead with the new exchange request for the desk lamp.

**assistant**:{'name': 'exchange_delivered_order_items', 'arguments': {"order_id": "#W6390527", "item_ids": ["8384507844"], "new_item_ids": ["7453605304"], "payment_method_id": "paypal_7644869"}}

**tool**:Error: non-delivered order cannot be exchanged

**assistant**:It seems there is an issue with the order status that prevents initiating a new exchange request. I will need to investigate further to resolve this.

Would you like me to transfer you to a human agent for assistance with this issue?

**user**:Yes, that would be helpful. Please transfer me to a human agent.

**assistant**:{'name': 'transfer_to_human_agents', 'arguments': {"summary": "The user wants to exchange only the desk lamp from order #W6390527, but the system is preventing the exchange due to an issue with the order status. The agent tried to cancel the previous exchange request but was unsuccessful because the order is not pending. The user has requested to be transferred to a human agent for further assistance."}}

**tool**:Transfer successful

Correct Trajectory after MUA-RL:

**user**:Hi! I'd like to exchange a couple of items I bought recently.

**assistant**:I'd be happy to help you exchange items from your recent purchase. To get started, I'll need to authenticate your account and look up your order details.

Could you please provide either your email address or your name along with your zip code? This will help me locate your user account and check your order history.

**user**:Sure! My name is Mei Kovacs and my zip code is 28236.

**assistant**:{'name': 'find_user_id_by_name_zip', 'arguments': {"first_name": "Mei", "last_name": "Kovacs", "zip": "28236"}}

**tool**:mei_kovacs_8020

**assistant**:{'name': 'get_user_details', 'arguments': {"user_id": "mei_kovacs_8020"}}

**tool**:{"name": {"first_name": "Mei", "last_name": "Kovacs"}, "address": {"address1": "317 Elm Street", "address2": "Suite 461", "city": "Charlotte", "country": "USA", "state": "NC", "zip": "28236"}, "email": "mei.kovacs8232@example.com", "payment_methods": {"paypal_7644869": {"source": "paypal", "id": "paypal_7644869"}}, "orders": ["#W6390527", "#W7800651", "#W8065207"]}

**assistant**:{'name': 'get_order_details', 'arguments': {"order_id": "#W6390527"}}

**tool**:{"order_id": "#W6390527", "user_id": "mei_kovacs_8020", "address": {"address1": "317 Elm Street", "address2": "Suite 461", "city": "Charlotte", "country": "USA", "state": "NC", "zip": "28236"}, "items": [{"name": "Desk Lamp", "product_id": "6817146515", "item_id": "8384507844", "price": 137.94, "options": {"color": "white", "brightness": "medium", "power source": "USB"}}, {"name": "Hiking Boots", "product_id": "7363354090", "item_id": "1615379700", "price": 253.89, "options": {"size": "10", "material": "synthetic", "waterproof": "yes"}}, {"name": "Water Bottle", "product_id": "8310926033", "item_id": "8538875209", "price": 45.13, "options": {"capacity": "500ml", "material": "glass", "color": "black"}}], "fulfillments": [{"tracking_id": ["745984883162"], "item_ids": ["8384507844", "1615379700", "8538875209"]}], "status": "delivered", "payment_history": [{"transaction_type": "payment", "amount": 436.96, "payment_method_id": "paypal_7644869"}]}

**assistant**:{'name': 'get_order_details', 'arguments': {"order_id": "#W7800651"}}

**tool**:{"order_id": "#W7800651", "user_id": "mei_kovacs_8020", "address": {"address1": "576 Oak Street", "address2": "Suite 970", "city": "San Francisco", "country": "USA", "state": "CA", "zip": "94137"}, "items": [{"name": "Gaming Mouse", "product_id": "5713490933", "item_id": "5796612084", "price": 158.89, "options": {"color": "RGB", "sensor type": "optical", "connectivity": "wired"}}, {"name": "Bookshelf", "product_id": "8600330539", "item_id": "3778705663", "price": 473.48, "options": {"material": "metal", "color": "black", "height": "6 ft"}}, {"name": "Office Chair", "product_id": "4794339885", "item_id": "4648362606", "price": 503.76, "options": {"material": "leather", "color": "black", "armrest": "adjustable", "backrest height": "high-back"}}], "fulfillments": [], "status": "pending", "payment_history": [{"transaction_type": "payment", "amount": 1136.13, "payment_method_id": "paypal_7644869"}]}

**assistant**:{'name': 'get_order_details', 'arguments': {"order_id": "#W8065207"}}

**tool**:{"order_id": "#W8065207", "user_id": "mei_kovacs_8020", "address": {"address1": "317 Elm Street", "address2": "Suite 461", "city": "Charlotte", "country": "USA", "state": "NC", "zip": "28236"}, "items": [{"name": "Smart Watch", "product_id": "6945232052", "item_id": "5694328282", "price": 323.19, "options": {"color": "gold", "band material": "leather", "display": "AMOLED"}}, {"name": "Smartphone", "product_id": "1801728040", "item_id": "1631373418", "price": 1291.21, "options": {"color": "gold", "storage": "128GB", "RAM": "6GB", "screen size": "6.1-inch"}}, {"name": "Luggage Set", "product_id": "5426915165", "item_id": "9956648681", "price": 452.62, "options": {"piece count": "4-piece", "color": "red", "material": "hardshell"}}, {"name": "Garden Hose", "product_id": "6679515468", "item_id": "4024196380", "price": 102.9, "options": {"length": "50ft", "material": "latex", "color": "black"}}], "fulfillments": [{"tracking_id": ["731287459054"], "item_ids": ["5694328282", "1631373418", "9956648681", "4024196380"]}], "status": "delivered", "payment_history": [{"transaction_type": "payment", "amount": 2169.92, "payment_method_id": "paypal_7644869"}]}]}

**assistant**:Thank you for providing your information, Mei. I've found your account and checked your order history. You have three orders:

1. Order #W6390527 (Delivered) 2. Order #W7800651 (Pending) 3. Order #W8065207 (Delivered)

Which order would you like to exchange items from? Also, could you let me know which items you'd like to exchange and what you'd like to exchange them for?

**user**:I'm not sure which order it was, but I want to exchange my water bottle for a bigger one, and my desk lamp for a less bright one.

**assistant**:I see. You'd like to exchange your water bottle for a bigger one and your desk lamp for a less bright one. Let me check which order these items are in.

Looking at your orders, I can see that both the water bottle and desk lamp are in your first order, #W6390527, which has been delivered.

For the water bottle, you currently have a 500ml glass bottle in black. For the desk lamp, you have a white one with medium brightness and USB power source.

To help you with the exchange, I'll need to: 1. Find suitable replacement items for both products 2. Check if they're available 3. Process the exchange for you

Would you like me to look up available options for both products?

**user**:Yes, please show me the options for both.

**assistant**:{'name': 'get_product_details', 'arguments': {"product_id": "8310926033"}}

**tool**:{"name": "Water Bottle", "product_id": "8310926033", "variants": {"1434748144": {"item_id": "1434748144", "options": {"capacity": "1000ml", "material": "glass", "color": "red"}, "available": false, "price": 49.72}, "4579334072": {"item_id": "4579334072", "options": {"capacity": "750ml", "material": "glass", "color": "black"}, "available": true, "price": 54.85}, "6469567736": {"item_id": "6469567736", "options": {"capacity": "1000ml", "material": "glass", "color": "blue"}, "available": false, "price": 47.84}, "3453331371": {"item_id": "3453331371", "options": {"capacity": "500ml", "material": "stainless steel", "color": "black"}, "available": true, "price": 52.79}, "2439754078": {"item_id": "2439754078", "options": {"capacity": "1000ml", "material": "stainless steel", "color": "red"}, "available": true, "price": 49.51}, "7843064651": {"item_id": "7843064651", "options": {"capacity": "750ml", "material": "stainless steel", "color": "blue"}, "available": true, "price": 50.14}, "7918497119": {"item_id": "7918497119", "options": {"capacity": "500ml", "material": "glass", "color": "blue"}, "available": false, "price": 54.51}, "5758737025": {"item_id": "5758737025", "options": {"capacity": "500ml", "material": "glass", "color": "green"}, "available": true, "price": 45.09}, "7533802601": {"item_id": "7533802601", "options": {"capacity": "500ml", "material": "stainless steel", "color": "green"}, "available": true, "price": 48.59}, "3229676465": {"item_id": "3229676465", "options": {"capacity": "500ml", "material": "plastic", "color": "black"}, "available": true, "price": 51.94}, "2366567022": {"item_id": "2366567022", "options": {"capacity": "1000ml", "material": "stainless

steel", "color": "blue"}, "available": false, "price": 54.04}, "6974536207": {"item_id": "6974536207", "options": {"capacity": "750ml", "material": "plastic", "color": "blue"}, "available": true, "price": 49.3}, "6777246137": {"item_id": "6777246137", "options": {"capacity": "750ml", "material": "stainless steel", "color": "red"}, "available": true, "price": 47.76}, "8538875209": {"item_id": "8538875209", "options": {"capacity": "500ml", "material": "glass", "color": "black"}, "available": true, "price": 45.13}, "9127591879": {"item_id": "9127591879", "options": {"capacity": "750ml", "material": "stainless steel", "color": "black"}, "available": false, "price": 48.47}, "7661609223": {"item_id": "7661609223", "options": {"capacity": "1000ml", "material": "stainless steel", "color": "black"}, "available": true, "price": 46.51}, "4947921075": {"item_id": "4947921075", "options": {"capacity": "750ml", "material": "stainless steel", "color": "green"}, "available": false, "price": 49.57}, "7199146548": {"item_id": "7199146548", "options": {"capacity": "750ml", "material": "plastic", "color": "black"}, "available": true, "price": 48.02}}}

**assistant**:{'name': 'get_product_details', 'arguments': {"product_id": "6817146515"}}

**tool**:{"name": "Desk Lamp", "product_id": "6817146515", "variants": {"9083642334": {"item_id": "9083642334", "options": {"color": "white", "brightness": "high", "power source": "USB"}, "available": true, "price": 164.28}, "4385534692": {"item_id": "4385534692", "options": {"color": "white", "brightness": "high", "power source": "AC adapter"}, "available": false, "price": 138.07}, "7624783998": {"item_id": "7624783998", "options": {"color": "black", "brightness": "high", "power source": "AC adapter"}, "available": true, "price": 154.17}, "1270145486": {"item_id": "1270145486", "options": {"color": "white", "brightness": "high", "power source": "battery"}, "available": false, "price": 144.07}, "5320792178": {"item_id": "5320792178", "options": {"color": "black", "brightness": "medium", "power source": "AC adapter"}, "available": true, "price": 135.24}, "5370728469": {"item_id": "5370728469", "options": {"color": "silver", "brightness": "medium", "power source": "USB"}, "available": true, "price": 164.97}, "6805564527": {"item_id": "6805564527", "options": {"color": "black", "brightness": "medium", "power source": "USB"}, "available": true, "price": 158.41}, "1569765161": {"item_id": "1569765161", "options": {"color": "silver", "brightness": "low", "power source": "AC adapter"}, "available": true, "price": 143.02}, "7453605304": {"item_id": "7453605304", "options": {"color": "silver", "brightness": "low", "power source": "battery"}, "available": true, "price": 150.01}, "9190635437": {"item_id": "9190635437", "options": {"color": "black", "brightness": "low", "power source": "USB"}, "available": true, "price": 153.23}, "4447749792": {"item_id": "4447749792", "options": {"color": "white", "brightness": "medium", "power source": "AC adapter"}, "available": false, "price": 139.8}, "8384507844": {"item_id": "8384507844", "options": {"color": "white", "brightness": "medium", "power source": "USB"}, "available": false, "price": 137.94}}}

**assistant**:Here are the available options for both products:

**Water Bottle (Product ID: 8310926033)** - 750ml glass, black - $54.85 (Available) - 500ml stainless steel, black - $52.79 (Available) - 1000ml stainless steel, red - $49.51 (Available) - 750ml stainless steel, red - $47.76 (Available) - 1000ml stainless steel, black - $46.51 (Available) - 500ml glass, green - $45.09 (Available) - 750ml plastic, black - $48.02 (Available) - 750ml plastic, blue - $48.30 (Available)

**Desk Lamp (Product ID: 6817146515)** - Black, medium brightness, USB - $158.41 (Available) - Black, medium brightness, AC adapter - $135.24 (Available) - Silver, medium brightness, USB - $164.97 (Available) - Silver, low brightness, AC adapter - $143.02 (Available) - Silver, low brightness, battery - $150.01 (Available) - Black, low brightness, USB - $153.23 (Available)

For your water bottle, you might want to consider the 750ml or 1000ml options. For your desk lamp, there are several medium or low brightness options available in different colors and power sources.

Which specific items would you like to exchange your current items for?

**user**:For the water bottle, I'd like the 1000ml stainless steel in black. For the desk lamp, I'd prefer the silver, low brightness, battery-powered one.

**assistant**:I can help you exchange your items. Let me summarize what you want to exchange:

1. Water Bottle: - Current item: 500ml glass, black (Item ID: 8538875209) - New item: 1000ml stainless steel, black (Item ID: 7661609223)

2. Desk Lamp: - Current item: White, medium brightness, USB (Item ID: 8384507844) - New item: Silver, low brightness, battery (Item ID: 7453605304)

Both items are from order #W6390527, which has been delivered.

To process this exchange, I'll need to use your payment method. You currently have one payment method on file: PayPal (ID: paypal_7644869).

There will be a price difference for both items: - Water Bottle: New price $46.51 vs old price $45.13 (difference: +$1.38) - Desk Lamp: New price $150.01 vs old price $137.94 (difference: +$12.07) Total difference: +$13.45

Would you like me to proceed with the exchange using your PayPal account? Please confirm with a "yes" if you'd like to continue.

**user**:I'd like to just exchange the desk lamp for now.

**assistant**:I understand you'd like to exchange just the desk lamp for now. Let me confirm the details:

- Current item: White, medium brightness, USB desk lamp (Item ID: 8384507844) - New item: Silver, low brightness, battery-powered desk lamp (Item ID: 7453605304) - Price difference: $150.01 (new) vs $137.94 (old) = +$12.07 - Payment method: PayPal (ID: paypal_7644869)

Would you like me to proceed with exchanging just the desk lamp? Please confirm with a "yes" if you'd like to continue.

**user**:Yes, please go ahead and exchange the desk lamp.

**assistant**:{'name': 'exchange_delivered_order_items', 'arguments': {"order_id": "#W6390527", "item_ids": ["8384507844"], "new_item_ids": ["7453605304"], "payment_method_id": "paypal_7644869"}}

**tool**:{"order_id": "#W6390527", "user_id": "mei_kovacs_8020", "address": {"address1": "317 Elm Street", "address2": "Suite 461", "city": "Charlotte", "country": "USA", "state": "NC", "zip": "28236"}, "items": [{"name": "Desk Lamp", "product_id": "6817146515", "item_id": "8384507844", "price": 137.94, "options": {"color": "white", "brightness": "medium", "power source": "USB"}}, {"name": "Hiking Boots", "product_id": "7363354090", "item_id": "1615379700", "price": 253.89, "options": {"size": "10", "material": "synthetic", "waterproof": "yes"}}, {"name": "Water Bottle", "product_id": "8310926033", "item_id": "8538875209", "price": 45.13, "options": {"capacity": "500ml", "material": "glass", "color": "black"}}], "fulfillments": [{"tracking_id": ["745984883162"], "item_ids": ["8384507844", "1615379700", "8538875209"]}], "status": "exchange requested", "payment_history": [{"transaction_type": "payment", "amount": 436.96, "payment_method_id": "paypal_7644869"}], "exchange_items": ["8384507844"], "exchange_new_items": ["7453605304"], "exchange_payment_method_id": "paypal_7644869", "exchange_price_difference": 12.07}

**assistant**:Great! I've successfully processed the exchange for your desk lamp. Here are the details:

- Your white, medium brightness, USB desk lamp (Item ID: 8384507844) has been exchanged for the silver, low brightness, battery-powered desk lamp (Item ID: 7453605304). - The price difference of $12.07 has been charged to your PayPal account. - Your order status has been updated to "exchange requested."

You will receive an email with instructions on how to return your old desk lamp and when to expect your new one.

Would you like to proceed with exchanging your water bottle as well, or is there anything else I can help you with?

**user**:No, that's all for now. Thank you! ###STOP###

