# OpenReview forum: "MUA-RL: Multi-turn User-interacting Agent Reinforcement Learning for agentic tool use"
_ICLR.cc/2026/Conference — ICLR 2026 Conference Withdrawn Submission_

### Official Review · Reviewer_DZFv · 2025-10-30

**Soundness:** 2
**Presentation:** 3
**Contribution:** 2
**Rating:** 2
**Confidence:** 4

**Summary:**

The paper introduces MUA-RL, a reinforcement learning framework designed for multi-turn, user-interactive agentic tool use. The authors conduct experiments demonstrating that, after reinforcement learning training, MUA-RL-32B achieves improved performance across several benchmarks.

**Strengths:**

1. The focus on multi-turn and user-interactive settings addresses essential aspects for developing more autonomous agents, introducing greater complexity to the learning process.

2. The paper includes evaluations on multiple benchmarks, such as $\tau^2$-Bench, BFCL-V3 Multi-Turn, and ACEBench.

**Weaknesses:**

1. Although the paper claims that MUA-RL represents a novel reinforcement learning framework, its novelty is not convincingly demonstrated. The task formulation, reward design, and training methodology appear relatively conventional, and the distinctions from existing approaches (such as [1,2,3]) are not clearly presented.

2. The $\tau$-bench  are used as both the training and test dataset.

3. The paper introduces two agentic data synthesis pipelines, but their effectiveness is not validated through ablation studies, leaving uncertainty about their contributions.

[1] WebGPT: Browser-assisted question-answering with human feedback.

[2] AGILE: A Novel Reinforcement Learning Framework of LLM Agents.

[3] Multi-turn Reinforcement Learning from Preference Human Feedback.

**Questions:**

1. If it is possible to obtain real trajectories using the MCP server, what is the motivation for generating LLM-simulated tool execution? What advantages do synthetic trajectories offer compared to real trajectories obtained from the MCP server?

---

### Official Review · Reviewer_8aYg · 2025-10-31

**Soundness:** 3
**Presentation:** 2
**Contribution:** 1
**Rating:** 2
**Confidence:** 3

**Summary:**

This paper presents an on-policy reinforcement learning loop that incorporates tool-use and interaction. Using their method named MUA-RL, the authors train a 32B model to outperform or match Deepseek and a large Qwen model.

**Strengths:**

1. Integration with MCP tool server with reinforcement learning and user interaction
2. Creating an RL loop on three interesting interactive tasks with a user: Tao bench, Berkeley Function-Calling Leaderboard, ACEBench Agent
3. Performs an ablation comparing non-thinking and cold start
4. The paper ablates different user LLMs that have a different LLM backend

**Weaknesses:**

1. The paper states that it introduces “a novel multi-turn user-interacting reinforcement learning framework that incorporates LLM-simulated users into the reinforcement learning rollouts.” (line 74-75) However, similar ideas have been explored in prior work. For example:
   a. LMRL-Gym [1]: Presents a framework for developing and evaluating human simulators, and integrates them into the LLM and RL   training loop
   b. Offline RL with Simulated Users [2,3]: Previous studies have incorporated user simulators into offline RL settings, including for interactive coding tasks
   c. SOTOPIA-RL [4]: Demonstrates related approaches for simulating human interactions in reinforcement learning contexts.
2. Concern about long term impact and framing: the paper frames as the primary contribution as placing tool use and user simulation in the same RL loop. I don’t see this as fundamentally different from those that were studied in the above papers and as such don’t see why this work should

[1] Abdulhai, Marwa, et al. "Lmrl gym: Benchmarks for multi-turn reinforcement learning with language models." arXiv preprint arXiv:2311.18232 (2023).
[2] Hong, Joey, Sergey Levine, and Anca Dragan. "Zero-shot goal-directed dialogue via rl on imagined conversations." arXiv preprint arXiv:2311.05584 (2023).
[3] Zhou, Yifei, et al. "Sweet-rl: Training multi-turn llm agents on collaborative reasoning tasks." arXiv preprint arXiv:2503.15478 (2025).
[4] Yu, Haofei, et al. "Sotopia-RL: Reward Design for Social Intelligence." arXiv preprint arXiv:2508.03905 (2025).

**Questions:**

- What is the significant novelty of including tool use in addition to a user centric approach?
- When doing the synthetic data generation pipeline for creating a dataset for further fine-tuning before RL fine-tuning, do the authors observe a difference between using only the LLM generated tool call responses versus the ones where the response comes from a genuine MCP server?

---

### Official Review · Reviewer_jMed · 2025-11-01

**Soundness:** 3
**Presentation:** 3
**Contribution:** 2
**Rating:** 6
**Confidence:** 2

**Summary:**

This paper proposes MUA-RL, a reinforcement learning framework that incorporates LLM-simulated users into RL rollouts for multi-turn agentic tool use. For cold-start SFT training, the authors construct two data synthesis pipelines, i.e., using LLM-simulated and real MCP server tool responses. For the RL process, the authors conduct a detailed analysis of training dynamics. MUA-RL-32B matches or outperforms much larger models like DeepSeek-V3-0324 and Qwen3-235B-A22B across multiple benchmarks, demonstrating strong performance.

**Strengths:**

1. This work proposes a framework to integrate LLM-simulated users into RL rollouts for agentic tool use, addressing the critical gap of dynamic user interactions in existing RL methods.
2. MUA-RL-32B matches or outperforms much larger models (DeepSeek-V3-0324, Qwen3-235B-A22B) across multiple benchmarks, demonstrating remarkable efficiency gains.
3. The paper provides a detailed analysis of the training dynamics. This analysis is insightful and better demonstrates the MUA-RL process.

**Weaknesses:**

1. This work only uses retail and airline datasets from TAU1-Bench for RL training, which is in a similar distribution as the testing environment, potentially limiting generalization to other domains.
2. This proposed method relies on GPT-4o as the user simulator during training. This is costly and may not be scalable.

**Questions:**

1. How does the framework prevent LLM-simulated users from generating unreasonable behaviors during training? What quality control mechanisms ensure the simulated users provide realistic interactions that won't lead to overfitting on artificial patterns?
2. Why was the non-thinking version of the model adopted instead of the thinking version?
3. What is the cost of LLM invocation when generating the cold-start data and in the RL process for simulation?

---

### Official Review · Reviewer_A4Zo · 2025-11-11

**Soundness:** 3
**Presentation:** 3
**Contribution:** 3
**Rating:** 4
**Confidence:** 3

**Summary:**

The paper proposes MUA-RL, a reinforcement learning (RL) framework that enables large language models (LLMs) to learn agentic tool use through multi-turn user interactions. The authors argue that existing reinforcement learning (RL) approaches for tool use lack the integration of genuinely dynamic users during the RL training process, and that this limits an agent’s ability to adapt to real-world conversations. Hence, the paper integrates LLM-simulated users into the reinforcement learning loop, enabling the agent to iteratively refine its understanding of user intent across multiple conversational turns. The method consists of a cold-start SFT phase, using two agentic data synthesis pipelines for high-quality cold-start: one with LLM-simulated tool responses, and another with real MCP server tool responses.

The authors report that MUA-RL-32B outperforms or matches much larger open-source models such as DeepSeek-V3-0324 and Qwen3-235B-A22B, and even rivals GPT-4o and GPT-4.1 on multiple tool-use benchmarks.

**Strengths:**

- The motivation is very compelling. Specifically, the authors note that users often adjust their questions and expectations based on the model’s responses
- The paper moves beyond static datasets or scripted dialogues and frames this problem of agent and user co-evolution
- The method MUA-RL combines synthetic cold-start data generation with interactive RL training, producing a more realistic environment for the agent.
- The evaluation spans four major multi-turn benchmarks—TAU1-Bench, TAU2-Bench, BFCL-V3 Multi Turn, and ACEBench Agent
- The paper provides analysis of training dynamics

**Weaknesses:**

- The paper uses simulated users as a proxy for human behavior, which is reasonable, but lacks empirical validation with actual human users. This could cause overfitting to synthetic conversational patterns. I suggest the authors do a user study with real human users to benchmark their method, as the title of the paper is also specifically addressing users. There needs to be validation of the LLM-simulated users as well.
- There is no discussion of the safety risks of the method and how it affects real users

**Questions:**

- The paper discusses using approximately 1600 trajectories across 9 scenarios for the cold-start training. More details about domain coverage and diversity would strengthen reproducibility claims.
- Integrating large user simulators like GPT-4o-2024-11-20 into the RL loop can be quite expensive. The authors do not provide estimates of training compute or efficiency.
- The reward is defined as binary. Did the authors experiment with other reward structures?

**Details Of Ethics Concerns:**

There needs to be a discussion of safety concerns regarding the method and its effect on users

---

### Note · Authors · 2026-01-25

I have read and agree with the venue's withdrawal policy on behalf of myself and my co-authors.